# The Impact of Fentanyl and Morphine on Maternal Hemodynamics in Spinal Anesthesia for Cesarean Section

**DOI:** 10.3390/ph18030392

**Published:** 2025-03-11

**Authors:** Ramona Celia Moisa, Nicoleta Negrut, Iulia Codruta Macovei, Cezar Cristian Mihai Moisa, Harrie Toms John, Paula Marian

**Affiliations:** 1Clinic of Anaesthesia and Intensive Care, Pelican Clinic, Medicover Hospital, 4104869 Oradea, Romania; moisa.ramona.celia@didactic.uoradea.ro; 2Doctoral School of Biomedical Sciences, University of Oradea, 410087 Oradea, Romania; 3Department of Surgery, Faculty of Medicine and Pharmacy, University of Oradea, 410087 Oradea, Romania; cmacovei@uoradea.ro; 4Department of Psycho-Neuroscience and Recovery, Faculty of Medicine and Pharmacy, University of Oradea, 410073 Oradea, Romania; 5Faculty of Medicine and Pharmacy, University of Oradea, 410087 Oradea, Romania; moisa.cezarcristianmihai@student.uoradea.ro; 6Department of Intensive Care, Epsom and St. Helier University Hospitals National Health Service Trust, Wrythe Lane, Carshalton, London SM5 1AA, UK; 7Department of Medical Disciplines, Faculty of Medicine and Pharmacy, University of Oradea, 410073 Oradea, Romania; paula.marian85@gmail.com

**Keywords:** spinal anesthesia, cesarean section, fentanyl, morphine, hemodynamics, hypotension, bradycardia

## Abstract

**Background:** Spinal anesthesia is considered the method of choice for elective cesarean sections; however, it is not without maternal–fetal risks. **Materials and Methods:** This study compared the effects on maternal hemodynamics of intrathecal administration of fentanyl or morphine in parturients undergoing spinal anesthesia with 0.5% hyperbaric bupivacaine, with doses varied between 7.5 and 11 mg, depending on the patient’s height. Data from a cohort of 170 parturients were analyzed. The administered doses were intrathecal morphine at 0.1 mL (100 µg, solution of 1 mg/mL) or fentanyl at 0.25 mL (25 µg, solution of 50 µg/mL). This study included 80 patients in the fentanyl (F) group and 90 in the morphine (M) group. **Results:** Group F showed significantly higher post-intervention systolic blood pressure values than group M (95.30 ± 12.99 mmHg vs. 90.58 ± 14.75 mmHg, *p* = 0.032). The incidence of vomiting was significantly less frequent in group F compared to group M (1, 1.3% vs. 10, 11.1%, *p* = 0.011). The total dose of ephedrine required for hypotension correction was significantly lower in the F group (12.75 ± 13.26 mg vs. 17.72 ± 16.73 mg, *p* = 0.035). **Conclusions:** The addition of fentanyl as an adjuvant alongside the local anesthetic in cesarean section is associated with enhanced hemodynamic stability compared to morphine, requiring lower doses of ephedrine and contributing to increased patient safety during elective cesarean surgery.

## 1. Introduction

Spinal anesthesia is considered the method of choice for elective cesarean sections around the world [1], being an easy-to-perform, fast-acting type of anesthesia that provides greater maternal and fetal safety than general anesthesia. Compared to general anesthesia, with airway manipulation and the risk of failed intubation and aspiration complications, spinal anesthesia avoids these complications and also reduces maternal and fetal exposure to intravenous anesthetic agents. Moreover, spinal anesthesia enables the mother to stay awake during childbirth, allowing an instant bonding experience with the newborn. These advantages have solidified its role as the gold standard for cesarean delivery [2].

Spinal anesthesia offers a multitude of benefits, but it also introduces various potential complications. Among the numerous potential adverse effects, maternal hypotension emerges as the most critical and recurrent issue because it occurs in approximately 70–80% of cases [3]. The manifestation of maternal symptoms such as nausea and vomiting together with dizziness results from severe hypotension, which reaches its most extreme form by causing loss of consciousness. An urgent medical issue arises when uteroplacental blood flow reduction occurs, which subsequently causes fetal acidosis along with neonatal distress [4]. However, there is no universally accepted definition of hypotension in the scientific literature. Some studies define it as a systolic blood pressure decrease of more than 20% from baseline, while others use an absolute threshold of less than 90 mmHg [4,5]. Regardless of the definition used, post-spinal hypotension remains a significant clinical challenge, necessitating effective preventive and management strategies [6].

The medical practice of decreasing local anesthetic dosage while adding opioid adjuvants has become increasingly popular as a technique to attain sufficient anesthesia levels without causing hemodynamic instability. Choosing an appropriate opioid adjuvant (morphine or fentanyl) emerges as a critical factor in altering hemodynamic responses during spinal anesthesia administration [7].

Fentanyl and morphine exhibit distinct pharmacokinetic and pharmacodynamic properties, influencing maternal hemodynamic parameters differently. Fentanyl, a lipophilic opioid, provides clinical advantages such as rapid onset and minimal rostral spread; however, it is associated with the risk of sudden respiratory depression and muscle rigidity [8,9,10]. On the other hand, morphine, a hydrophilic opioid with a prolonged analgesic effect, is a valuable option for postoperative pain management but may induce severe respiratory depression or arrest and hemodynamic instability (hypotension and bradycardia) in the first 24 h, with a biphasic pattern and significant rostral spread [9,11]. The intricate examination and comprehension of these differences constitute essential elements in enhancing the selection process of opioids for spinal anesthesia during cesarean sections [8].

The precise impact of these opioids on maternal blood pressure and cardiovascular function remains undefined, representing a significant gap in existing research.

The frequent occurrence of maternal hypotension combined with its potential adverse effects on both mother and fetus necessitates a direct comparative study of intrathecal F versus morphine during spinal anesthesia for cesarean delivery. We hypothesize that fentanyl and morphine, due to their distinct pharmacokinetic and pharmacodynamic properties, will have different effects on maternal hemodynamics. A thorough examination of how these agents affect maternal blood pressure alongside vasopressor needs and hemodynamic stability will yield essential insights to refine anesthetic techniques.

## 2. Results

This study included 175 patients. Five cases were excluded. The rest were divided into two groups based on the intrathecal opioid adjuvant used: 80 patients received fentanyl (Group F), and 90 patients received morphine (Group M). The baseline characteristics of the two groups are summarized in Table 1. There were no significant differences between the groups regarding demographic and clinical characteristics (*p* > 0.05 for all). The SMD was close to 0 for most parameters, with a maximum of 0.1, indicating minimal imbalance that is unlikely to affect the comparability of the groups.

### 2.1. Blood Pressure and Ventricular Rate

After the intervention, a significant difference in BP_2S_ was observed between the groups, with the F group showing significantly higher values compared to the M group (95.30 ± 12.99 mmHg vs. 90.58 ± 14.75 mmHg, *p* = 0.032). No statistically significant differences were recorded for BP_2D_ and MAP_2_ values between the F group and the M group (55.26 ± 11.05 vs. 53.08 ± 9.87, *p* = 0.284, and 68.61 ± 11.21 vs. 65.58 ± 10.69, *p* = 0.121, respectively), as shown in Figure 1.

The type of anesthesia did not statistically significantly influence the changes in blood pressure between the time before and after anesthesia administration, as shown in Table 2.

The incidence of nausea did not show a statistically significant difference between the groups. In group F, 12 patients (15.0%) reported nausea, compared to 17 patients (18.9%) in group M (χ^2^ = 0.453, *p* = 0.501), indicating a similar distribution of nausea incidence between the two groups.

In group F, only 1 patient (1.3%) experienced vomiting, compared to 10 patients (11.1%) in group M. Fisher’s exact test revealed a statistically significant difference (*p* = 0.011).

No statistically significant differences were identified in the VR_5_ values recorded five minutes after the intervention between the two groups (M: 85.57 ± 22.99 bpm vs. F: 88.26 ± 22.82 bpm; *p* = 0.424), as shown in Figure 2. The data were non-normally distributed (M, *p* < 0.001; F, *p* < 0.001), and the Mann–Whitney U test was used to evaluate differences between the groups.

### 2.2. Peripheral Capillary Oxygen Saturation

For both groups, the differences between SpO_2_ values before and those 5 min after the intervention were evaluated using the Wilcoxon signed-rank test. The results indicated a statistically significant difference between them for group M (98.49 ± 0.70% vs. 97.28 ± 1.17%, Z = −6.919, *p* < 0.001) and group F(98.49 ± 0.76% vs. 97.34 ± 1.10%, Z = −6.497, *p* < 0.001). The medians of the differences suggest a consistent change in SpO_2_ values between the two evaluation moments, as shown in Figure 3. The differences in SpO_2_ reduction between the groups were compared using an independent-sample *t*-test. The mean reduction in group M was −10.38 ± 22.15%, while, in group F, it was −7.25 ± 22.78%. Levene’s test indicated homogeneity of variances (F = 0.515, *p* = 0.425). The comparison of means did not reveal a statistically significant difference between the groups (t = −0.907, *p* = 0.366, mean difference = −3.13, 95% CI: −9.94 ÷ 3.68). The effect size was small, as indicated by Cohen’s d (d = −0.139). These results suggest that the type of anesthesia did not significantly influence the reduction in SpO_2_.

### 2.3. Administration of Ephedrine and Atropine Based on the Type of Anesthetic Used

The proportion of patients requiring ephedrine did not differ significantly between group M (67, 74.44%) and group F (53, 66.25%), according to the chi-square test (*p* = 0.242). The ephedrine dose was significantly lower in patients from group F (12.75 ± 13.26 mg) compared to group M (17.72 ± 16.73 mg), as indicated by the *t*-test [t(168) = 2.129, *p* = 0.035)] and as shown in Figure 4. The data distribution was normal in both groups, and Levene’s test confirmed the homogeneity of variances (*p* = 0.106).

There was no statistically significant difference in the number of patients who received atropine based on the type of anesthesia (group M: 9, 10% vs. F: 3, 3.75%, *p* = 0.198).

## 3. Discussion

Studying the different effects of fentanyl and morphine on maternal hemodynamics in the context of cesarean section with spinal anesthesia is essential for understanding their impact on cardiovascular stability.

### 3.1. Blood Pressure and Ventricular Rate

The medical literature shows that sympathetic blockade is the main cause of maternal hypotension after spinal anesthesia [10]. This blockade triggers vasodilation along with a reduced venous return and subsequently lowers cardiac output. The selection of opioid adjuvants is critical for controlling the cardiovascular system’s reactions.

In this study, fentanyl was associated with significantly higher post-intervention BP_S_ than morphine (95.30 ± 12.99 mmHg vs. 90.58 ± 14.75 mmHg, *p* = 0.032). However, no significant differences were observed in BP_D_ or MAP between the two groups. These differences are based on the distinct pharmacodynamics of the two opioids. Fentanyl has a rapid onset of action, which may lead to a quick inhibition of sympathetic tone, explaining the higher blood pressure values [12]. On the other hand, morphine has a slower onset and a prolonged duration of action, which may contribute to more persistent hypotensive effects through prolonged activity on the vasomotor centers of the brainstem and the release of histamine from mast cells, which is responsible for peripheral vasodilation [13]. To the best of our knowledge, no studies have been reported to directly analyze this aspect, which adds originality and relevance to our investigation. However, further studies are needed to monitor BP variations over a longer period.

Nausea and vomiting are common side effects of spinal anesthesia, often triggered by hypotension-induced cerebral hypoperfusion and direct opioid effects on the chemoreceptor trigger zone [14]. In the present study, vomiting was significantly less frequent in the F group (1.3%) than in the M group (11.1%, *p* = 0.011), whereas nausea rates were similar. Our results are consistent with the medical literature. Research by DeSousa et al. (2014) supports that intrathecal morphine administration leads to an increased frequency of nausea and vomiting [15]. In a study conducted by Weigls et al. (2017) on two groups of parturients undergoing spinal anesthesia with fentanyl (28) and fentanyl with morphine (30), the incidence of dyspeptic syndrome (nausea and vomiting) was statistically significantly higher in the group that also received morphine (1 vs. 11, *p* < 0.001) [16]. The hydrophilic characteristics of this substance allow it to move cephalad through cerebrospinal fluid, leading to greater potential for nausea and vomiting. Due to its lipophilic properties, fentanyl reduces the rostral spread and reduces the risk of opioid-induced nausea and vomiting [17]. The properties of fentanyl may contribute to maternal comfort during cesarean delivery; however, its use should be carefully considered based on individual patient characteristics and potential risks, such as respiratory depression and muscle rigidity.

Bradyarrhythmia after spinal anesthesia happens mainly through reduced venous return, decreased heart output, and increased parasympathetic response [18]. The autonomic effects of opioid adjuvants can modulate this response along with heart rate changes. In the current study, no significant differences in VR were observed between the F and M groups five minutes after spinal anesthesia administration. This result contrasts with findings from Sibanyoni et al. (2022), who suggested that intrathecal morphine may be more likely to induce bradycardia due to its prolonged effects on central autonomic regulation [19]. However, the lack of significant differences in the current study may be due to the relatively low opioid doses used and the effective management of hemodynamic changes with ephedrine and atropine.

### 3.2. Peripheral Capillary Oxygen Saturation

Opioid adjuvants in spinal anesthesia can impact oxygenation by modulating central respiratory control, particularly through the depression of brainstem respiratory centers [20].

In the present study, both fentanyl and morphine led to slight but statistically significant reductions in SpO_2_ values five minutes after spinal anesthesia administration, though no significant differences were observed between the two groups. The current study findings match those presented by Sibanyoni et al. (2022), who documented morphine’s extended reaction with central opioid receptors leading to delayed respiratory depression [19]. The observed decreases in oxygen saturation monitored in the present research were minimal, thus demonstrating that both substances can be safely employed during spinal anesthesia without causing significant respiratory issues.

### 3.3. Administration of Ephedrine and Atropine Based on the Type of Anesthetic Used

Spinal anesthesia frequently causes hypotensive effects that need vasopressor therapy to maintain stable BP. The mixed α- and β-adrenergic agonist ephedrine serves as the primary medication to treat spinal anesthesia-induced hypotension because it enhances cardiac output and systemic vascular resistance [5]. The administration of atropine occurs as a therapeutic measure for treating clinically significant bradycardia that develops after spinal anesthesia procedures.

In the current study, the administration frequency of ephedrine remained similar between patients who received fentanyl compared to those who received morphine (66.25% vs. 74.44%, *p* = 0.242). This research showed that the F group needed lower ephedrine doses than the M group for hypotension correction procedures (*p* = 0.035). Fentanyl recipients received 12.75 ± 13.26 mg of ephedrine, while morphine subjects received 17.72 ± 16.73 mg. Fentanyl appears to be associated with a reduced need for vasopressor intervention, likely due to its pharmacokinetic properties. However, these findings should be interpreted cautiously, as hemodynamic responses may vary depending on individual patient factors and anesthesia protocols.

The study findings support previous work by Ebrie et al. (2022), who demonstrated that fentanyl acts as a spinal anesthesia adjuvant to reduce local anesthetic doses, thereby limiting sympathetic blockade [7]. Studies by Gallagher et al. (2024) demonstrated that fentanyl administered intravenously leads to better BP control than morphine due to its quick elimination from systemic circulation and small effects on central hemodynamic management [21]. However, there are no studies investigating the effects of intrathecal administration of these opioids on BP.

The increased ephedrine requirement in the morphine group can be attributed to morphine’s prolonged central nervous system effects, particularly its interaction with brainstem vasomotor centers, which exacerbates hypotension and necessitates greater vasopressor support [9]. Morphine’s hydrophilic nature also leads to prolonged spinal fluid circulation, potentially extending its depressant effects on cardiovascular function [9].

Regarding atropine administration, no statistically significant differences were observed between the groups (M: 10% vs. F: 3.75%, *p* = 0.198). This suggests that, while both opioids may induce some degree of bradycardia, the incidence was relatively low and did not vary significantly between the groups. These results are consistent with findings by Shahid et al. (2024), who reported that intrathecal opioids rarely cause severe bradycardia, necessitating atropine intervention, especially at low doses [2].

These findings support the preferential use of fentanyl as an adjuvant in spinal anesthesia for cesarean sections due to its association with lower vasopressor requirements and improved hemodynamic stability.

### 3.4. Study Limitations and Clinical Implications

This study provides data on maternal BP responses to intrathecal administration of fentanyl and morphine; however, certain limitations exist. This research was conducted at a single center, which may limit the generalizability of the results. Additionally, this study did not assess pruritus, a known side effect of intrathecal opioid administration. Further validation through multisite trials is necessary to confirm these findings, and pruritus should be included to provide a more comprehensive evaluation of adverse effects.

The clinical data indicate that fentanyl is associated with more stable maternal hemodynamics, a lower need for vasopressors, and a reduced incidence of vomiting compared to morphine. These findings may influence anesthetic protocols, leading to a greater preference for fentanyl, especially for patients who need enhanced hemodynamic control or risk developing hypotension.

## 4. Materials and Methods

### 4.1. Study Design

This retrospective study was conducted on records of parturients admitted for delivery at Pelican Clinic, Medicover Hospital, Romania, between 1 January 2023 and 30 June 2024.

It received approval from the Pelican Clinic Ethics Committee of Medicover Hospital (Approval No. 2421/10.12.2019) and adhered to the ethical principles of the World Medical Association’s Declaration of Helsinki (2024). Informed consent was obtained from all participants at the time of their admission to the hospital.

Pre-established eligibility criteria determined the subjects’ inclusion in this study. Clinical data analysis was conducted based on this study’s inclusion and exclusion criteria without active allocation interventions, Figure 5. Data extraction followed a standardized protocol to ensure objectivity and minimize selection bias, and patient information was anonymized before statistical analysis.

The primary outcome of this study was maternal hemodynamic stability, assessed through changes in blood pressure (BP) following spinal anesthesia administration. Secondary outcomes included the following:Vasopressor requirement (total dose of ephedrine administered for BP correction).Anticholinergic requirement for bradycardia correction (total dose of atropine administered).Incidence of adverse effects (nausea and vomiting).Ventricular rate (VR) variability after anesthesia.Peripheral capillary oxygen saturation (SpO_2_) changes after anesthesia.

Potential confounders, including maternal age, BMI, baseline BP, bupivacaine dose, number of previous cesarean sections, and presence of uterine contractions, were considered as they could influence the choice of opioid and maternal hemodynamic outcomes.

### 4.2. Inclusion Criteria

American Society of Anesthesiologists (ASA) classification II;Age between 18 and 50 years;No significant medical history, including cardiovascular, respiratory, neurological, endocrine, or hematological disorders;No known drug or food allergies;No chronic medication use, including antihypertensive drugs or treatment for any condition, particularly preeclampsia;No history of chronic pain or regular use of analgesics (opioids or non-opioids)No contraindications to spinal anesthesia, such as coagulopathy, local infection, or anatomical abnormalities;Body weight over 50 kg;Singleton, viable fetus confirmed via ultrasound;Gestational age of at least 37 weeks at the time of elective cesarean section;Indication for elective cesarean section, excluding emergency procedures or cases with maternal or fetal distress.

### 4.3. Exclusion Criteria

Acute or chronic fetal distress, diagnosed pre- or postnatally;Fetal malformations, detected intrauterine or postpartum;Development of preeclampsia during pregnancy or peripartum;Need for surgical reintervention within 72 h postpartum, regardless of the cause;Requirement for opioid administration or other intravenous anesthetics during the procedure;Spinal anesthesia failure, necessitating conversion to general anesthesia;Absence of essential data in the patient observation chart;Blood loss of more than 500 mL;Sensory block maximum at T4 level;Transferred to another hospital.

### 4.4. Anesthetic Protocol and Monitoring

All patients underwent spinal anesthesia following the hospital’s standardized anesthesia protocol. Preoperatively, all patients included in this study were monitored using the Dräger Infinity Delta system (2001, Lübeck, Germany), which allowed for noninvasive recording of vital parameters. BP, VR, and SpO_2_ were continuously measured to ensure patient safety. Premedication consisted of 500 mL of Ringer’s lactate, 40 mg pantoprazole, and 10 mg metoclopramide, administered intravenously.

Spinal anesthesia was performed under strict aseptic conditions with the patient in a sitting position. The procedure was conducted at the L3–L4 intervertebral space using a 27-gauge pencil-point spinal needle with an introducer. After local anesthetic infiltration with 2 mL of 1% lidocaine, the intrathecal space was identified, and a single-shot anesthetic mixture was administered.

All patients received hyperbaric bupivacaine 0.5% (5 mg/mL) in doses ranging from 7.5 mg (1.5 mL) to 11 mg (2.2 mL), adjusted according to the patient’s height, administered intrathecally, combined with either intrathecal morphine at a dose of 0.1 mL (100 µg, solution of 1 mg/mL) or intrathecal fentanyl at a dose of 0.25 mL (25 µg, solution of 50 µg/mL). The doses of intrathecal fentanyl (25 µg) and morphine (100 µg) were determined based on the hospital’s standardized anesthesia protocol for cesarean section.

Following the spinal injection, the patient was immediately placed in the supine position with a right hip wedge to prevent inferior vena cava compression. Non-invasive BP monitoring was performed. Additional monitored parameters included VR and SpO_2_. The motor and sensory block levels were assessed, with the motor block evaluated using the Bromage scale, where 4 indicated maximum efficacy and the sensory block was considered maximum at the T4 level.

Oxygen was administered at a flow rate of 3–4 L/min when oxygen saturation dropped below 95%. Hypotension was managed with ephedrine, prepared in a dilution of 5 mg/mL. If an initial drop in BP was detected, an ephedrine dose of 5–15 mg was administered, depending on the severity of hypotension. Ephedrine administration was repeated to restore optimal systolic blood pressure (BP_S_). Bradycardia, defined as a heart rate below 55 beats per minute (bpm) within the first 5 min, was treated with an intravenous dose of 0.5 mg atropine.

Ringer’s lactate (10–20 mL/kg/h) was administered to maintain adequate intravascular volume. After fetal extraction, 4 mg dexamethasone and 4 mg ondansetron were administered to prevent postoperative nausea and vomiting.

### 4.5. Data Collection

The collected data included demographic and lifestyle characteristics such as age, place of residence, and smoking status. Clinical and anthropometric parameters were recorded, including weight, height, body mass index (BMI), number of pregnancies and births, history of cesarean delivery, presence of contractions at the time of anesthesia, and previous analgesia.

Hemodynamic parameters were monitored before (_1_) and after anesthesia (_2_), including BP_S_, diastolic blood pressure (BP_D_), mean arterial pressure (MAP), VR, and SpO_2_.

Blood pressure was measured before (_1_) and every minute after spinal anesthesia until 3 min after fetal extraction, with the lowest recorded value being considered for this study (_2_).

VR was measured before (_1_) and after the intervention during the first 5 min, considering the lowest recorded values (_2_).

SpO_2_ was measured before anesthesia (_1_) and 5 min after its administration (_2_).

Nausea and vomiting were recorded until fetal extraction.

This study is part of a larger research project analyzing maternal and neonatal physiological parameters. A complementary study focusing on neonatal outcomes is being published separately.

### 4.6. Statistical Analysis

The collected data were analyzed using the IBM SPSS Statistics software (version 29.0.2.0 (20)). The demographic and clinical characteristics of the subjects in the two study groups were evaluated using descriptive statistics.

The sample size of patients included in this study was calculated based on the total number of cesarean deliveries recorded during the analyzed period in the hospital.

The following variables were considered:p—the probability of the occurrence of the studied phenomenon (0 ≤ p ≤ 1);q—the complementary probability, q = 1 − p;t—the probability factor;Δx—the allowable margin of error;N—the total population (the total number of cesarean deliveries during the analyzed period).

To determine the minimum required number of patients for this study (n), we used the following formula: n = t^2^pq/(Δx^2^ + t^2^pq/N).

For a robust estimation, we considered the most unfavorable case, where p = q = 0.5, which maximizes the value of n. For a 95% probability, the t factor corresponds to a value of 1.96, and the allowable margin of error was set at 0.1. In this study, 735 patients underwent cesarean sections during the analyzed period. Applying the formula, this study’s minimum required sample size is 85 patients.

Parametric and non-parametric tests were applied for group comparisons depending on the data distribution.

The Shapiro–Wilk test was used to assess the normality of distribution for continuous variables. Subsequently, the comparison tests were adapted to each dataset. For variables with a normal distribution, the independent t-test was applied, while, for those with a non-normal distribution, the Mann–Whitney U test was used. Differences between values measured before and after the intervention were analyzed using the Wilcoxon Signed-Rank test.

The Chi-square test was used to compare the distribution of categorical data, and Fisher’s exact test was applied when cell frequencies were low.

We calculated the standardized mean difference (SMD) for each variable to assess the balance of baseline characteristics between the two groups. Standard calculation formulas were used for both continuous and categorical variables. An SMD below 0.1 was considered negligible for interpretation, indicating well-balanced groups. An SMD between 0.1 and 0.2 suggested a minor imbalance, which may still be acceptable. However, an SMD of 0.2 or higher indicated a significant imbalance.

All statistical tests were conducted using a significance threshold of *p* < 0.05, and the Levene test was employed to assess the homogeneity of variances.

## 5. Conclusions

The current study suggests that intrathecal fentanyl was associated with BP stability in spinal anesthesia for cesarean section. Fentanyl administration was linked to higher BP values and a reduced need for vasopressor medication and antiemetic interventions compared to morphine. However, given the retrospective nature of this study, further prospective research is needed to confirm these findings and assess potential confounding factors.

Additional studies should investigate neonatal outcomes, as well as long-term maternal recovery, to provide a more comprehensive understanding of the hemodynamic effects of different opioids in spinal anesthesia for cesarean delivery.

## Figures and Tables

**Figure 1 pharmaceuticals-18-00392-f001:**
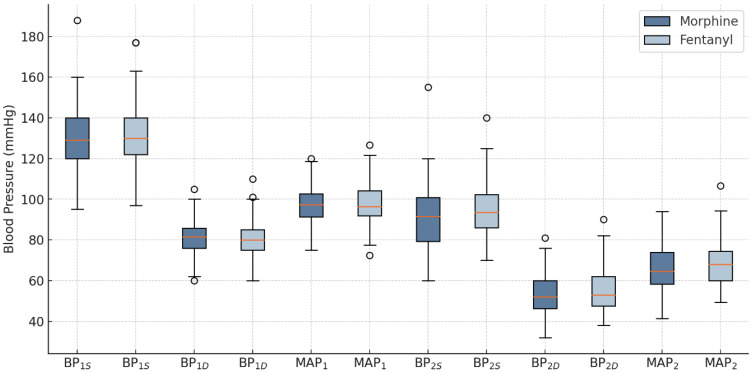
Blood pressure changes in groups. BP_S_—systolic blood pressure; BP_D_—diastolic blood pressure; MAP—mean arterial pressure; _1_—before the intervention; _2_—after the intervention; °—outliers.

**Figure 2 pharmaceuticals-18-00392-f002:**
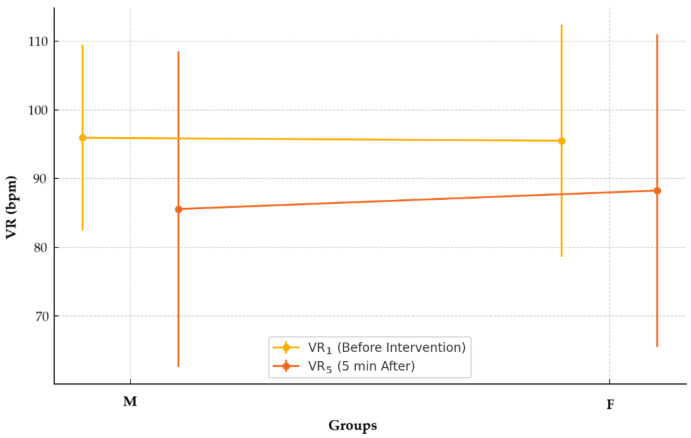
Ventricular rates before and after intervention in the anesthetic group.

**Figure 3 pharmaceuticals-18-00392-f003:**
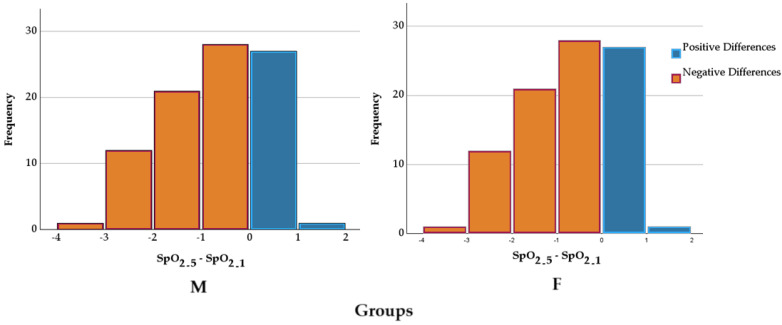
SpO_2_ differences in groups according to anesthesia. SpO_2_—peripheral capillary oxygen saturation; _1_—before the intervention; _5_—5 min after the intervention.

**Figure 4 pharmaceuticals-18-00392-f004:**
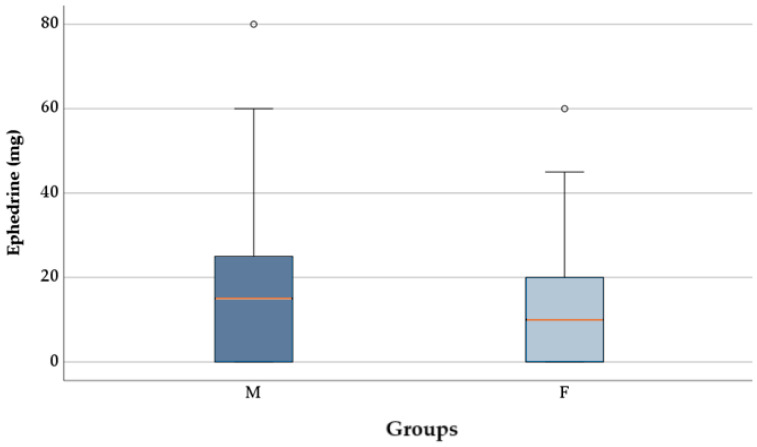
Ephedrine dosages between groups. 

—mean; °—outliers.

**Figure 5 pharmaceuticals-18-00392-f005:**
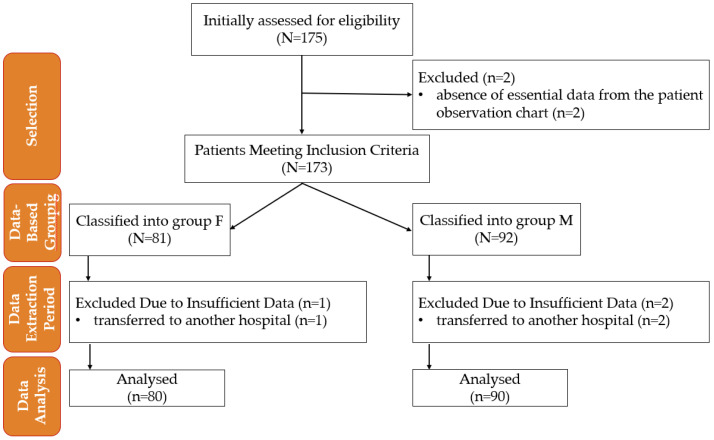
Flowchart of patient inclusion and data analysis in this study. M—morphine; F—fentanyl; n—number.

**Table 1 pharmaceuticals-18-00392-t001:** Characteristics of the study groups.

Parameter	Group F (n = 80)	Group M (n = 90)	*p*-Value	SMD
DD				
Age, years, M ± SD (min, max)	31.61 ± 4.37 (23.50)	32.17 ± 4.38 (22.44)	0.434 *	0.1
Urban residence, n (%)	55 (68.8)	62 (68.9)	0.984 **	0.0
Smoker, n (%)	13 (14.44)	15 18.75)	0.450 **	0.0
Clinical and laboratory data				
Weight, kg, M ± SD (min, max)	77 ± 12.83 (50, 110)	77.84 ± 15.05 (53, 131)	0.805 *	0.0
Height, cm, M ± SD (min, max)	164.35 ± 5.86 (150, 180)	165.21 ± 6.31 (150, 178)	0.945 *	0.0
BMI, M ± SD (min, max)	28.25 ± 3.55 (19.20, 39.06)	28.82 ± 5.29 (19.47, 44.37)	0.384 *	0.0
Number of pregnancies, n (%)				
0	42 (52.5)	50 (55.6)	0.471 ***	0.0
1	28 (35.0)	32 (35.6)		0.0
2	7 (8.8)	3 (3.3)		0.1
3	3 (3.8)	5 (5.6)		0.0
Number of births, n (%)				
0	51 (63.8)	55 (61.1)	0.402 ***	0.0
1	25 (31.2)	30 (33.3)		0.0
2	4 (5.0)	3 (3.3)		0.0
3	0 (0.0)	2 (2.2)		0.0
PCS, n (%)	26 (32.5)	27 (30.0)	0.725 **	0.0
Contractions present, n (%)	21 (26.2)	16 (17.8)	0.181 **	0.1
Previous analgesia, n (%)	4 (5.0)	1 (1.1)	0.189 ****	0.1
BP_1S_, mmHg, M ± SD	132.09 ± 14.85	130.42 ± 14.51	0.475 **	0.1
BP_1D_, mmHg, M ± SD	80.50 ± 9.09	80.97 ± 8.62	0.446 **	0.0
MAP_1_, mmHg, M ± SD	97.70 ± 9.72	97.45 ± 9.63	0.989 **	0.0
VR_1_, bpm, M ± SD	95.51 ± 16.89	95.94 ± 13.54	0.854 °	0.0
SpO_2___1_, %, M ± SD	98.49 ± 0.76	98.49 ± 0.70	0.934 *	0.0
Bupivacaine, mg, M ± SD	9.84 ± 0.75	9.96 ± 0.79	0.342 °	0.1

DD—demographic data; BP_1S_—systolic blood pressure, before the intervention; BP_1D_—diastolic blood pressure, before the intervention; MAP—mean arterial pressure; VR—ventricular; PCS—previous cesarean sections; SpO_2_—peripheral capillary oxygen saturation; _1_—before anesthesia; n—number; M—mean; SD—standard deviation; min—minimum; max—maximum; *—Testul Mann-Whitney U; **—Pearson Chi-Square; ***—Likelihood Ratio; ****—Fisher’s Exact Test; °—*t*-test; SMD—standardized mean difference.

**Table 2 pharmaceuticals-18-00392-t002:** Variation of blood pressure according to the type of anesthesia.

Parameter	Group F	Group M	*p*-Value
ΔBP_S_, mmHg, M ± SD	−36.79 ± 17.64	−39.84 ± 18.88	0.278 *
ΔBP_D_, mmHg, M ± SD	−25.24 ± 13.29	−27.89 ± 11.73	0.168 **
ΔMAP, mmHg, M ± SD	−29.09 ± 13.40	−31.87 ± 13.08	0.172 *

BP_S_—systolic blood pressure; BP_D_—diastolic blood pressure; MAP—mean arterial pressure; Δ—the difference in blood pressure between the initially recorded value (_1_) and the final recorded value (_2_); M—mean; SD—standard deviation; **—*t*-test; *—Testul Mann-Whitney U.

## Data Availability

The data presented in this study are available on request from the corresponding author. The data are not publicly available due to the data are part of an ongoing study.

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
