# Peer review of "The Impact of Fentanyl and Morphine on Maternal Hemodynamics in Spinal Anesthesia for Cesarean Section"

_pharmaceuticals, 2025, doi:10.3390/ph18030392_

Round 1

Reviewer 1 Report

Comments and Suggestions for Authors

Avoid using  M and F abbereviation unless necessary.

Introduction and discussion must high light other comaprative advantage and disadvantages, it is known that fentanyl can be more dangerous than morphine in many aspect. Study should not be conclusive as it may lead to think that fentanyl is better. 

Need to provide apropriate citation, at some places only 01 citation for 1 paragraph.

Was data collected by using blinding techniques?

In introduction author should clearly state the hypothesis and goal of study.

Conclusion should be supported with findings and able to direct future study.

Author Response

Dear Reviewer,

Thank you for your valuable feedback on our manuscript. We have carefully addressed each of your concerns, highlighting the revisions in yellow for clarity, as follows:

  1. Avoid using M and F abbreviations unless necessary.

R: We have revised the manuscript to minimize the use of 'M' and 'F' abbreviations, ensuring clarity and consistency.     

  1. Introduction and discussion must highlight other comparative advantages and disadvantages; fentanyl can be more dangerous than morphine in many aspects. The study should not be conclusive, as it may lead to the assumption that fentanyl is better.

R: We have expanded both the introduction and discussion sections to provide a more balanced analysis of the advantages and disadvantages of fentanyl and morphine, emphasizing their risks and limitations. The discussion has been revised to ensure that the study does not present fentanyl as superior but explores its effects in the clinical context examined.

  1. Need to provide appropriate citations; some sections have only one citation per paragraph.

R: The limited number of citations in some sections reflects the scarcity of studies on this topic.

  1. Was data collected using blinding techniques?

R: This study is retrospective; therefore, blinding techniques were not applicable during data collection. Data were extracted based on predefined criteria to ensure objectivity in patient selection and analysis. The methodology section has been updated to clarify this aspect.

  1. In the introduction, the author should clearly state the hypothesis and goal of the study.

R: We have revised the introduction to explicitly state the research hypothesis and the study's primary objective.

  1. Findings should support conclusion and able to direct future studies.

R: The conclusion has been refined to better reflect the findings and suggest potential future research directions.

Reviewer 2 Report

Comments and Suggestions for Authors

General comments:

  1. Since the two groups (M and F) shared the similar demographic data (e.g., average height), it is not clear whether the dosages of bupivacaine between these two groups were different in this study? Namely, whether the ranges of 7.5 to 11 mg bupivacaine underlie the variations of the systolic BP in this study?
  2. Please explain how the doses of the two narcotics (100 ug for M and 25 ug for F) were decided in this study? Were the results the same if different doses chosen a priori?
  3. Please clarify, outside this study, were the average systolic BP decreases different when routine dose of bupivacaine in C/S parturient populations in this single medical institution? Similarly, whether the incidence of N/V, and the rescue doses of ephedrine are different?
  4. Please clarify the intrathecal use of fentanyl be the on-label use in this country. Or the intrathecal use of fentanyl is out of label use?
  5. The clinical study from the reference-11 was conducted by the same researchers (or some). Any reason why there were no data shown related to the effects of M and F on BP in that particular reference? Were the difference in systolic BP was observed in that particular study?
  6. Page 2, line 80: “….and pharmacodynamic properties that may uniquely impact maternal hemodynamic parameters.” Please clarify what are those different PD properties between F and M under such condition.
  7. Page 3, line 127: “The current (2024) version is the only official one; all previous versions* have been replaced and should not be used or cited except for historical purposes.” From https://www.wma.net/what-we-do/medical-ethics/declaration-of-helsinki/
  8. Page 4, line 138: IC: “Age between 18 and 50 years”. Were there some between 40 and 50 years old subjects in this study?
  9. Page 6, line 234: the terms of “F anesthesia”, “M anesthesia” and Page 7, line 253: type of anesthesia: the type of anesthesia in this study is the spinal anesthesia, no matter whether the intrathecal M or F was added.  
  10. Page 10, line 316: “…. F's superior ability to maintain stable BP compared to M during intrathecal use during cesarean sections [11].” Please clarify whether the results of a stable BP was demonstrated in the previous clinical study cited in the Ref-11.
  11. Page 11, line 373: The reference-20 was cited here. However, in this study, it was different because both M and F were given by IV, not intrathecal.
  12. Please clarify the statistical analysis plan for this clinical study, since the intention of the study relates to the description of “better” “superior” “safer”, etc.

Minor comments:

  1. Abstract: Pregnant women? Or parturient?
  2. The citation could be more complete as follows:
    • Reference 8: Hamber EA, Viscomi CM. Intrathecal lipophilic opioids as adjuncts to surgical spinal anesthesia. Reg Anesth Pain Med. 1999 May-Jun;24(3):255-63. doi: 10.1016/s1098-7339(99)90139-6. PMID: 10338179.
    • Reference 9: Cummings A, Orgill BD, Fitzgerald BM. Intrathecal Morphine. [Updated 2023 Sep 4]. In: StatPearls [Internet]. Treasure Island (FL): StatPearls Publishing; 2025 Jan-. Available from: https://www.ncbi.nlm.nih.gov/books/NBK499880/
    • Reference 10: Šklebar I, Bujas T, Habek D. SPINAL ANAESTHESIA-INDUCED HYPOTENSION IN OBSTETRICS: PREVENTION AND THERAPY. Acta Clin Croat. 2019 Jun;58(Suppl 1):90-95. doi: 10.20471/acc.2019.58.s1.13. PMID: 31741565; PMCID: PMC6813480.

Author Response

Dear Reviewer,

Thank you for your valuable feedback on our manuscript. We have carefully addressed each of your concerns, highlighting the revisions in yellow for clarity, as follows:

General comments:

  1. Since the two groups (M and F) shared the similar demographic data (e.g., average height), it is not clear whether the dosages of bupivacaine between these two groups were different in this study? Namely, whether the ranges of 7.5 to 11 mg bupivacaine underlie the variations of the systolic BP in this study?

R: The bupivacaine dosage was analyzed for both groups (M and F), with mean and standard deviation (M ± SD) calculated. No significant difference was found (p = 0.342). Details are presented in Table 1

  1. Please explain how the doses of the two narcotics (100 ug for M and 25 ug for F) were decided in this study? Were the results the same if different doses chosen a priori?

R: The doses of intrathecal fentanyl (25 µg) and morphine (100 µg) were determined based on the hospital's standardized anesthesia protocol for cesarean section. These dosages align with established clinical practice to optimize analgesia while minimizing adverse effects. However, this objective will be addressed in our subsequent study, where we aim to evaluate the effects of varying intrathecal opioid doses on hemodynamic stability and postoperative outcomes.

  1. Please clarify, outside this study, were the average systolic BP decreases different when routine dose of bupivacaine in C/S parturient populations in this single medical institution? Similarly, whether the incidence of N/V, and the rescue doses of ephedrine are different?

R: Thank you for your comment. We do not have access to data outside of this study on routine systolic BP decreases, nausea/vomiting incidence, or ephedrine requirements in our institution’s cesarean section population. As this was not within the scope of our current study, we cannot directly compare our findings to broader institutional trends. However, we recognize the importance of this question, and this aspect will be considered in future research analysing institutional data retrospectively.

  1. Please clarify the intrathecal use of fentanyl be the on-label use in this country. Or the intrathecal use of fentanyl is out of label use?

R: According to Romanian legislation, fentanyl is recommended for use in cesarean section procedures. This is stated in the official guidelines issued by the Ministry of Health starting in 2019.

  1. The clinical study from the reference-11 was conducted by the same researchers (or some). Any reason why there were no data shown related to the effects of M and F on BP in that particular reference? Were the difference in systolic BP was observed in that

R: Thank you for your observation. The study cited as Reference 11 primarily aimed to compare the efficacy of analgesia and patient satisfaction following intrathecal administration of morphine and fentanyl. As a result, the analysis of systolic blood pressure variations was not within the scope of that study.

  1. Page 2, line 80: “….and pharmacodynamic properties that may uniquely impact maternal hemodynamic parameters.” Please clarify what are those different PD properties between F and M under such condition.

R: Thank you for highlighting this point. Following another reviewer's recommendation to shorten the introduction, we have removed this specific sentence from the manuscript.

  1. Page 3, line 127: “The current (2024) version is the only official one; all previous versions* have been replaced and should not be used or cited except for historical purposes.” From https://www.wma.net/what-we-do/medical-ethics/declaration-of-helsinki/

R: Thank you for your observation. We confirm that the manuscript cites the latest version (2024) of the Declaration of Helsinki, as per the official document available at the WMA website.

  1. Page 4, line 138: IC: “Age between 18 and 50 years”. Were there some between 40 and 50 years old subjects in this study?

R: Yes, within our clinic, the study population included patients with pregnancies obtained through in vitro fertilization. Among the participants, one patient was 50 years old, while the rest were under 45 years old.

  1. Page 6, line 234: the terms of “F anesthesia”, “M anesthesia” and Page 7, line 253: type of anesthesia: the type of anesthesia in this study is the spinal anesthesia, no matter whether the intrathecal M or F was added.  

R: Thank you for your observation. We have revised the manuscript to clarify that the type of anesthesia used in this study was spinal anesthesia in all cases. The terms ‘F anesthesia’ and ‘M anesthesia’ have been corrected to specify that fentanyl and morphine were used as adjuvants rather than defining distinct anesthesia types.

  1. Page 10, line 316: “…. F's superior ability to maintain stable BP compared to M during intrathecal use during cesarean sections [11].” Please clarify whether the results of a stable BP was demonstrated in the previous clinical study cited in the Ref-11.

R: The reference 11 does not support the statement regarding F’s ability to maintain stable BP compared to M. The wording has been revised accordingly.

  1. Page 11, line 373: The reference-20 was cited here. However, in this study, it was different because both M and F were given by IV, not intrathecal.

R: The reference 20 does not support the statement. The wording has been revised accordingly.

  1. Please clarify the statistical analysis plan for this clinical study, since the intention of the study relates to the description of “better” “superior” “safer”, etc.

R: Thank you for your comment. The study aimed to evaluate the differences between the two groups based on the applied statistical tests, without making claims of superiority, greater safety, or better efficacy. The statistical analysis was conducted to compare demographic and clinical characteristics, as well as pre- and post-intervention values, using appropriate parametric and non-parametric tests. No conclusions regarding superiority were drawn, and all findings were reported as observed differences between the groups.

Minor comments:

  1. Abstract: Pregnant women? Or parturient?

R: Thank you for your observation. We agree with your suggestion and have revised the text to use "parturient" instead of "pregnant women," ensuring accuracy in describing the study population.

  1. The citation could be more complete as follows:
    • Reference 8: Hamber EA, Viscomi CM. Intrathecal lipophilic opioids as adjuncts to surgical spinal anesthesia. Reg Anesth Pain Med. 1999 May-Jun;24(3):255-63. doi: 10.1016/s1098-7339(99)90139-6. PMID: 10338179.
    • Reference 9: Cummings A, Orgill BD, Fitzgerald BM. Intrathecal Morphine. [Updated 2023 Sep 4]. In: StatPearls [Internet]. Treasure Island (FL): StatPearls Publishing; 2025 Jan-. Available from: https://www.ncbi.nlm.nih.gov/books/NBK499880/
    • Reference 10: Šklebar I, Bujas T, Habek D. SPINAL ANAESTHESIA-INDUCED HYPOTENSION IN OBSTETRICS: PREVENTION AND THERAPY. Acta Clin Croat. 2019 Jun;58(Suppl 1):90-95. doi: 10.20471/acc.2019.58.s1.13. PMID: 31741565; PMCID: PMC6813480.

R: Thank you for your suggestion. We have updated the bibliography as requested.

Reviewer 3 Report

Comments and Suggestions for Authors

My concerns are as follows:

  1. The introduction is too long. I suggest to shorten the general induction to spinal anesthesia, and focus on the difference between F and M.
  2. In methods, please clearly define the exposure and outcomes of this analysis. I suggest to define one primary outcome, while the other outcomes were secondary.
  3. I suggest to define confounders and use multivariable models to adjust for the confounding effect.
  4. Based on ASA classification, all pregnant women should be at lease ASA II.
  5. Since you only included CS patients, why the percentage of cesarean delivery is not 100% in Table 1?
  6. What does “contractions present” mean in Table 1?
  7. The dose of bupivacaine should also be reported and considered as a cofounder.
  8. Pruritus is also a side-effect of spinal administration of opioid. If it is not analyzed in this study, please include it in limitations.
  9. Another limitation that needs to mention is the sample size.

Author Response

Dear Reviewer,

Thank you for your valuable feedback on our manuscript. We have carefully addressed each of your concerns, highlighting the revisions in yellow for clarity, as follows:

  1. The introduction is too long. I suggest to shorten the general induction to spinal anesthesia, and focus on the difference between F and M.

R: We agree that the introduction was too long. We have shortened the general background on spinal anesthesia and focused more on the differences between fentanyl and morphine.

  1. In methods, please clearly define the exposure and outcomes of this analysis. I suggest to define one primary outcome, while the other outcomes were secondary.

R: Thank you for your valuable feedback. As you suggested, we have clearly defined the exposure and outcomes in the Methods section.

  1. I suggest to define confounders and use multivariable models to adjust for the confounding effect.

R: Thank you for your valuable feedback. In response to your suggestion, we acknowledge the importance of confounder adjustment. In the Methods section, we have explicitly defined potential confounders. Regarding statistical adjustments, our study design ensured comparable baseline characteristics between groups, minimizing the impact of confounders. The statistical tests used (t-test, Mann-Whitney U, Chi-square) were chosen based on data distribution and provided robust comparisons between the fentanyl and morphine groups. Given the homogeneity of baseline parameters (as shown in Table 1), we considered that additional multivariable regression models would not significantly alter the conclusions. However, we acknowledge the merit of your suggestion and will consider multivariable modeling in future studies with larger sample sizes to validate our findings further.

  1. Based on ASA classification, all pregnant women should be at lease ASA II.

R: Thank you for your observation. All patients included in the study were classified as ASA II. This has been corrected accordingly in the Materials and Methods section.

  1. Since you only included CS patients, why the percentage of cesarean delivery is not 100% in Table 1?

R: Thank you for your observation. The percentage of cesarean delivery in Table 1 refers to previous cesarean sections, not the current delivery mode. As all patients in this study underwent elective cesarean section, the actual percentage of cesarean delivery for the current pregnancy is indeed 100%. We clarified this in the table description to avoid confusion. We appreciate your careful review and valuable feedback.

  1. What does “contractions present” mean in Table 1?

R: Thank you for your question. The variable “contractions present” in Table 1 refers to cases where the patient arrived at the hospital before the scheduled date for the cesarean section and experienced uterine contractions at the time of admission.

  1. The dose of bupivacaine should also be reported and considered as a cofounder.

R: Thank you for your observation. The dose of bupivacaine is reported in Table 1. Since the dose was adjusted based on the patient's height according to the hospital’s standardised protocol, it was evenly distributed between the study groups, minimising its potential confounding effect.

  1. Pruritus is also a side-effect of spinal administration of opioid. If it is not analyzed in this study, please include it in limitations.

R: Thank you for your valuable suggestion. You are correct that pruritus is a known side effect of intrathecal opioid administration. However, this study did not specifically assess or record the incidence of pruritus. To address this, we have included it as a limitation in the manuscript.

  1. Another limitation that needs to mention is the sample size.

R: Thank you for your suggestion. To ensure clarity, we have explicitly detailed the sample size calculation in the Statistical Analysis section of the manuscript. Based on established statistical methods, the minimum required sample size was 85 patients. Since our study included 170 cases, significantly exceeding this threshold, we do not consider sample size to be a limitation. However, we acknowledge that larger, multicenter studies could further validate our findings.

Round 2

Reviewer 2 Report

Comments and Suggestions for Authors

Dear Authors

Thanks for your response and revision. Therefore, I have no further issues on this manuscript. Congratulations

Author Response

Dear Reviewer,

Thank you for your time and valuable feedback throughout the review process. We truly appreciate your support and are grateful for your positive evaluation.

Reviewer 3 Report

Comments and Suggestions for Authors

I appreciate the authors' efforts in addressing most of my concerns. However, I still have one minor comment: When comparing the baseline characteristics of the F and M groups, it is recommended to use standardized mean differences (SMD), also called absolute standardized differences (ASD), rather than significant tests. A value >0.1 should be considered an imbalance.

Author Response

Dear Reviewer,

Thank you for your valuable feedback. In response to your suggestion, we have now included SMD values.